# Combining Graph Attention and Recurrent Neural Networks in a Variational Autoencoder for Molecular Representation Learning and Drug Design

**Alex T. Müller** [1]   **Kenneth Atz** [1]   **Michael Reutlinger** [1]   **Nicolas Zorn** [1]

## Abstract

Finding a meaningful molecular representation that can be leveraged for a variety of tasks in chemical sciences and drug discovery is of wide interest, and new representation learning techniques are continuously being explored. Here, we investigate the fusion of graph attention neural networks with recurrent neural networks within a variational autoencoder framework for molecular representation learning. This combination leverages the strengths of both architectures to capture properties of molecular structures, enabling more effective encoding and flexible decoding processes. With the resulting representation, we observe competitive performance in quantitative structure-activity relationship (QSAR) benchmarks, a high validity and drug-likeness of randomly sampled molecules and robustness for linear latent space interpolation between two molecules. Our approach holds promise for facilitating downstream tasks such as clustering, QSAR, virtual screening and generative molecular design, all unified in one molecular representation.

## 1. Introduction

In drug discovery, the goal is to find chemical structures with desired properties and experimental outcomes, such as a biological activity on a target protein of interest (Hughes et al., 2011). To navigate the vast chemical space, computer-assisted drug design approaches necessitate molecular representations that can be correlated with such outcomes (Schneider, 2018). More recent endeavors have focused on obtaining global models for drug discovery that do not only capture aspects of small molecules for single proteins, cell lines or diseases, but can be applied globally. Ultimately, there is interest to move from discovering to designing desired compounds from scratch (Schneider et al., 2020).

We are interested in finding a single molecular feature extraction method, also termed foundation model (Ahmad et al., 2022), to create a representation that can be used for multiple down-stream tasks such as quantitative structure–activity relationship (QSAR), virtual screening and *de novo* molecular design. The representation should cover the diverse aspects of chemical space, from molecular properties to structures, scaffolds and functional groups of small molecules. Instead of relying on human expert knowledge or rule-based feature engineering, representation learning with deep learning models has become a status quo in the field of molecular descriptors (Fabian et al., 2020; Duvenaud et al., 2015). Once trained on specific tasks relevant to drug discovery, the model's latent space representation can be used for QSAR modeling, virtual screening tasks, *de novo* design and to cluster physical samples in a high-throughput screening library.

### 1.1. Molecular Representations

A plethora of molecular descriptors exist as a way to represent chemical structures in numerical form (Todeschini & Consonni, 2008). However, we are hereafter focusing on representations that are learned by using neural networks.

Chemical language models are recurrent neural networks (RNNs) or transformers trained on string representations of molecules, such as simplified molecular-input line-entry system (SMILES) string (Weininger, 1988) and Self-Referencing Embedded Strings (SELFIES) (Krenn et al., 2020). Chemical language models have shown successful applications in reaction prediction (Schwaller et al., 2019), retrosynthesis planning (Segler et al., 2018), QSAR modeling and virtual screening (Muratov et al., 2020; van Tilborg et al., 2022) as well as in *de novo* molecular design (Gupta et al., 2018; Müller et al., 2018). A notable example for meaningful molecular representation learning is MolBERT, a bidirectional encoder representation from transformer (BERT) architecture with property prediction,

---

[1]Roche Pharma Research and Early Development (pRED), Roche Innovation Center Basel, F. Hoffmann-La Roche Ltd., Basel, Switzerland. Correspondence to: Alex T. Müller <alex.mueller@roche.com>.

*Accepted at the 1st Machine Learning for Life and Material Sciences Workshop at ICML 2024.*

string equivalence and language modeling heads (Fabian et al., 2020). Other more recent examples using the BERT architecture are ChemBERTa (Chithrananda et al., 2020) and ChemBERTa-2 (Ahmad et al., 2022), as well as MolFORMER (Ross et al., 2022). Wen *et al.* used transformers on unhashed extended-connectivity fingerprints (ECFP) (Rogers & Hahn, 2010) with radius one as inputs to train a BERT model, termed FP-BERT, in a self-supervised manner (Wen et al., 2022).

A challenge of string-based molecular representations is that one molecule can be represented by multiple different strings. This is overcome when molecules are represented as undirected graphs, fitting more naturally the connectivity of atoms and bonds. In graph representations, the molecular graph $\mathcal{G}$ consists of a set of vertices $\mathcal{V}$ and edges $\mathcal{E}$, *i.e.*, $\mathcal{G} = (\mathcal{V}, \mathcal{E})$. Vertices (*i.e.*, $v_i \in \mathcal{V}$) represent atoms, and whose edges (*i.e.*, $e_i \in \mathcal{E}$) constitute their bonds. Based on $\mathcal{G}$, graph neural networks (GNNs) (Kipf & Welling, 2016) can be used to learn molecular representations. Atz *et al.* provide a structured and harmonized overview of molecular geometric deep learning (Atz et al., 2021), and a more recent survey gives an overview on a large number of methods for graph-based molecular representation learning and related applications, categorized by input representation, algorithm, domain knowledge and task (Guo et al., 2023). As with transformers, applications of GNNs range from molecular representation learning (Duvenaud et al., 2015; Fang et al., 2022; Atz et al., 2024), QSAR (Kearnes et al., 2016), chemical reaction prediction (Nippa et al., 2024) to generative molecular design (Maziarz et al., 2021; Isert et al., 2023) and quantum property prediction by modeling a computationally expensive density functional theory (DFT) calculations (Gilmer et al., 2017; Atz et al., 2022).

A noteworthy example proposed a graph neural network with attention mechanisms at both the atom and molecule level for small molecule representation (called Attentive FP), which is able to learn both local and non-local properties of a two dimensional (2D) molecular graph (Xiong et al., 2019). The Attentive FP model was evaluated for acute toxicity prediction tasks, where it was identified as the best-performing model among five GNNs (Ketkar et al., 2023), and further showed competitive performance on proprietary ADME datasets (Broccatelli et al., 2022). Also, the presence of graph attention weights allows for visualization of atom importance for specific prediction tasks. Other evaluations of Attentive FP have shown its performance for drug-target interactions (Lei et al., 2022), LogD prediction (Duan et al., 2023) or to improve the performance of band gap approximation of organic materials (Khan et al., 2023).

Further examples that also include three-dimensional geometry information are spatial graph convolutional networks (SGCN) (Danel et al., 2020), directional message passing neural networks (DimeNet) (Gasteiger et al., 2020), heterogeneous molecular graph neural networks (HMGNN) (Shui & Karypis, 2020) and geometry-enhanced molecular representation learning (GEM) (Fang et al., 2022). In GEM, message passing is made sensitive to both topology and geometry, whereas (Zhu et al., 2022) unify 2D molecular graphs and 3D conformers for pre-training.

Regardless of the applied method, representation learning methods aim to transform a discrete representation, *i.e.*, a molecular graph or a SMILES-string, into a continuous descriptor space, where chemically similar molecules have similar representations. This allows to sample new molecules in close regions of chemical space, *e.g.* for hit expansion in drug discovery projects. The general advantages of continuous over discrete representations were already discussed in detail by (Gómez-Bombarelli et al., 2018).

## 1.2. Variational Autoencoders

Most of the aforementioned methods undergo self-supervised training or are designed as autoencoders. A challenge with autoencoders is that they tend to overfit and thereby create an irregular, non-continuous latent space. Different regularization approaches have been tried to obtain continuous latent representations. Most notably, a variational autoencoder (VAE) (Kingma & Welling, 2013), where not a single point in latent space is learned, but a probabilistic latent space with a distribution for each training example. One of the first examples for learning continuous molecular representations was to use VAEs trained on SMILES-strings (Gómez-Bombarelli et al., 2018). VAEs have also already been established for molecular graphs, where (Jin et al., 2018) used a junction tree VAE to incrementally create molecules, and (Maziarz et al., 2021) improved this approach by using structurally relevant motifs to ensure chemical validity. Jin *et al.* adequately summarized the advantage of a continuous representation obtained by VAEs as "learning to represent molecules in a continuous manner that facilitates the prediction and optimization of their properties (encoding); and learning to map an optimized continuous representation back into a molecular graph with improved properties (decoding)" (Jin et al., 2018).

Herein, we introduce a **G**raph **I**nfused **R**epresentation **A**ssembled **F**rom a multi-**F**aceted variational auto-**E**ncoder (GIRAFFE). GIRAFFE is a VAE model with a graph attention neural network (Xiong et al., 2019) as encoder and a RNN with LSTM cells (Hochreiter & Schmidhuber, 1997) as decoder. Even though graph-based models with sequential generation such as MoLeR would enjoy perfect validity of generated molecules (Maziarz et al., 2021), RNNs have

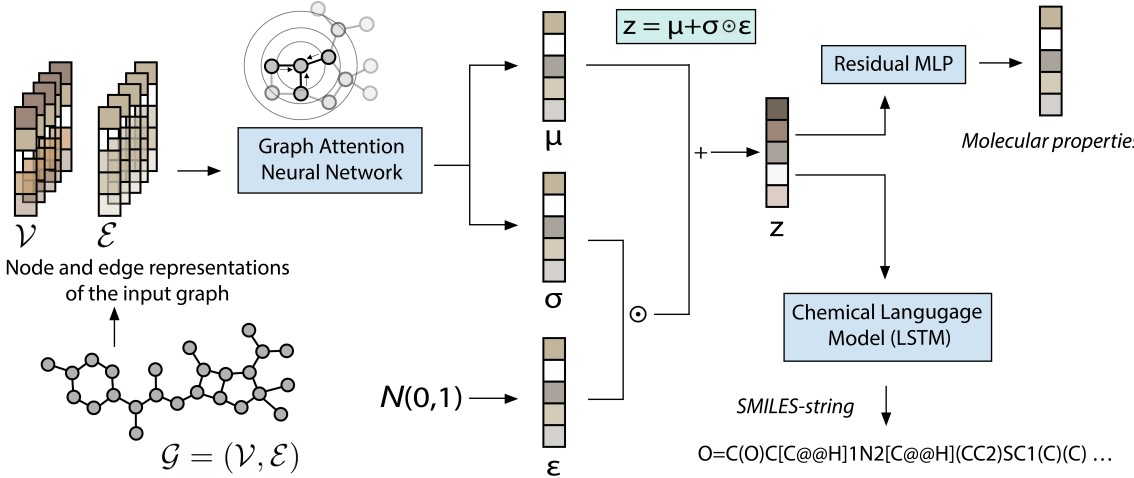

*Figure 1.* Neural network architecture of GIRAFFE. Molecules are represented as two-dimensional graphs, *i.e.*, $\mathcal{G} = (\mathcal{V}, \mathcal{E})$, where $\mathcal{V}$ and $\mathcal{E}$ are transformed using graph attention neural networks (*i.e.*, Attentive FP). As a result of the subsequent pooling process, mean $\mu$ and standard deviation $\sigma$ tensors are obtained, from which a latent space representation is sampled using the reparametrization trick with $\epsilon$ as the source of stochasticity. The resulting vector $z$ describes a condensed latent representation, which is fed to a multilayer perceptron (MLP) to predict calculated molecular properties of the molecule, and a LSTM to reconstruct a SMILES representation of the input graph.

shown high validity and successful applications in drug design (Merk et al., 2018), have a steerable curiosity component by using temperature and are not limited to previously observed motifs (Moret et al., 2023). In addition to the encoder and decoder models, and similar to (Winter et al., 2019), we include the prediction of RDKit descriptors (RDKit) from the latent space during training to improve the relevance of the learned representation for QSAR and help with latent space disentaglement. The training hence encompasses a translation task from molecular graphs to a compressed latent representation and from there back to SMILES-strings and calculable properties. We use Attentive FP (Xiong et al., 2019) on 2D graphs due to its proven performance in QSAR tasks and potential explainability. Finally, we attempt to disentangle and enforce a continuous latent representation by utilizing Kullback-Leibler Divergence (KLD) as a regularization term in a $\beta$-VAE loss setting. This should enforce the constraint that the learned latent variable distribution matches the standard normal distribution of the prior (Kingma & Welling, 2013).

## 2. Methods

### 2.1. Dataset

10M random molecules were extracted from PubChem (Kim et al., 2019) with valid SMILES-strings of maximum 128 characters. The dataset was randomly split into 9M molecules for training and 1M molecules for validation.

### 2.2. Model

GIRAFFE consists of three parts: (I) an attention-based graph neural network encoder, (II) a LSTM decoder and (III) a fully-connected multi-layer perceptron (MLP) regressor for property prediction (Figure 1). We use the Attentive FP implementation in PyTorch Geometric (Fey & Lenssen, 2019) with 2 layers, 2 time steps and 512 hidden dimensions as the encoder. As decoder, a 2-layer LSTM with 512 hidden dimensions and 64 token embedding dimensions is used, whereas the MLP for property prediction consisted of 2 fully connected layers with 512 hidden dimensions. Both LSTM and MLP are implemented in PyTorch (Ansel et al., 2024).

### 2.3. Training

We trained all parts of GIRAFFE end to end using the Adam optimizer (Kingma & Ba, 2014) with an initial learning rate of 0.001 and a step-wise decay of 0.75 every 10 epochs for a total of 150 epochs with 1000 steps per epoch. In each step, a batch of 256 molecules was randomly sampled from the available training pool. Molecules were represented as graphs $\mathcal{G} = (\mathcal{V}, \mathcal{E})$ using 32 node ($\mathcal{V}$) and 10 edge ($\mathcal{E}$) features adapted from (Xiong et al., 2019) as described in Table A.2. For each batch of molecules, a PyTorch Geometric (Fey & Lenssen, 2019) data object was constructed containing the graphs with node and edge features. During training, the graph is fed to the encoder, which produces a mean $\mu$ and standard deviation $\sigma$ (both with 512 dimensions) as output, from which a latent space representation is sampled in VAE-fashion using the reparametrization trick (Kingma & Welling, 2013). The resulting vector $z$ is used as the initial

*Table 1.* RMSE values for regression benchmarks. The best values per task are presented in bold. Values for the other representations were taken from (Wen et al., 2022) and (Fabian et al., 2020). *our models.

| DESCRIPTOR | ESOL | FREESOLV | LIPOP |
|---|---|---|---|
| RDKIT | 0.69 ± 0.08 | 1.67 ± 0.45 | 0.74 ± 0.04 |
| ECFP4 | 0.90 ± 0.06 | 2.88 ± 0.38 | 0.77 ± 0.03 |
| CDDD | 0.57 ± 0.06 | 1.46 ± 0.43 | 0.67 ± 0.02 |
| MOLBERT | **0.55 ± 0.07** | 1.52 ± 0.66 | **0.60 ± 0.01** |
| FP-BERT | 0.67 ± 0.07 | **1.07 ± 0.18** | 0.67 ± 0.02 |
| NONVAE* | 0.57 ± 0.07 | **1.07 ± 0.34** | 0.61 ± 0.01 |
| GIRAFFE* | **0.55 ± 0.08** | 1.11 ± 0.31 | 0.67 ± 0.03 |

*Table 2.* AUROC values for classification benchmarks. The best values per task are presented in bold. Values for the other representations were taken from (Wen et al., 2022) and (Fabian et al., 2020). *our models.

| DESCRIPTOR | BACE | BBBP | HIV |
|---|---|---|---|
| RDKIT | 0.83 ± 0.00 | 0.70 ± 0.00 | 0.71 ± 0.00 |
| ECFP4 | **0.85 ± 0.00** | 0.68 ± 0.00 | 0.71 ± 0.00 |
| CDDD | 0.83 ± 0.00 | **0.76 ± 0.00** | 0.75 ± 0.00 |
| MOLBERT | **0.85 ± 0.00** | 0.75 ± 0.00 | 0.75 ± 0.00 |
| FP-BERT | – | 0.71 ± 0.01 | **0.78 ± 0.01** |
| NONVAE* | **0.85 ± 0.00** | 0.72 ± 0.00 | 0.71 ± 0.00 |
| GIRAFFE* | **0.85 ± 0.00** | 0.71 ± 0.00 | 0.72 ± 0.00 |

hidden state for first layer of the decoder LSTM, which is trained to reconstruct the corresponding SMILES-string. SMILES-strings are recreated starting from a random atom in every training step. In parallel, the sampled latent space vector is fed to a fully-connected MLP regressor to predict all available RDKit descriptors (RDKit) for the given compound, normalized to $[0, 1]$. An overview of the process is provided in Figure 1. Training was stopped once the total validation loss increased. All models were trained on a single NVIDIA A100-SXM4-40GB GPU.

### 2.3.1. LOSS

We employ a standard VAE loss (Kingma & Welling, 2013) with the following modifications: The total training loss $\mathcal{L}$ (Eq. 1) is constructed from the SMILES categorical cross-entropy reconstruction error $\mathcal{L}_S$ of the decoder, the mean squared error $\mathcal{L}_P$ of the property prediction MLP as well as a KLD distance loss $\mathcal{L}_{KLD}$. $\mathcal{L}_P$ is weighted by a factor $\lambda_P = 10$, whereas $\mathcal{L}_{KLD}$ is weighted by a variable factor $\beta$. As a comparison, we also trained the same model architecture on the same data and using the same hyperparameters but without the loss term $\mathcal{L}_{KLD}$. We call the resulting model "nonVAE".

$$\mathcal{L} = \mathcal{L}_S + \lambda_P \times \mathcal{L}_P + \beta \times \mathcal{L}_{KLD} \quad (1)$$

### 2.3.2. $\beta$ ANNEALING

To achieve a stable training run, we utilized a cyclical KLD annealing technique for the factor $\beta$ of the term $\mathcal{L}_{KLD}$ to optimize our VAE. $\beta$ was gradually increased following a cyclical linear schedule to reach a maximum of 0.2 over 5 cycles, allowing the model to initially focus on the reconstruction losses $\mathcal{L}_S$ and $\mathcal{L}_P$ before progressively concentrating more on the KLD term. After this initial cyclical annealing, the cyclical linear schedule was continued until the end of the training (Fu et al., 2019). We investigated different annealing schedules with cycle sizes varying between 1000 and 20'000 steps, linear or sigmoid slopes and maximum values of 0.05 to 0.25.

### 2.4. Benchmark

We followed (Wen et al., 2022) and (Fabian et al., 2020) to benchmark the learned representation of our model using support vector machine models with the same hyperparameters. Results from previously published representations were taken directly from (Fabian et al., 2020) and (Wen et al., 2022) and not reproduced. For a fair comparison to our non-fine-tuned model, the results without fine-tuning were used for MolBERT (Fabian et al., 2020).

## 3. Results

For all performance assessments, we used the model checkpoint at the epoch which corresponded to the lowest total validation loss, as defined in Section 2.3.1. We evaluated different annealing strategies and found the following strategies performed similarly in terms of validation loss and equally well in the tested benchmarks: (I) cycles of 7'500 steps of linear increase followed by a plateau of 2'500 steps of constant values, with 4 growing cycles and a maximum $\beta$ of 0.2 (Figure A.5, top, red) achieved the lowest validation loss after 45'000 steps; and (II) cycles of 3'750 steps of sigmoidal increase followed by a plateau of 1'250 steps of constant values, with 20 growing cycles and a maximum $\beta$ of 0.2 (Figure A.5, bottom, blue).

### 3.1. QSAR Benchmark

Benchmark results are presented in Table 1 for regression tasks and in Table 2 for classification tasks of the Molecule Net benchmark (Wu et al., 2018). Both the GIRAFFE and the nonVAE model match the performance of most of the existing representations in several QSAR benchmarks.

### 3.2. Validity of Sampled SMILES

During training, the validity of the sampled SMILES-strings plateaued at around 96%. To investigate the advantage of a continuous latent space obtained by a VAE, we performed

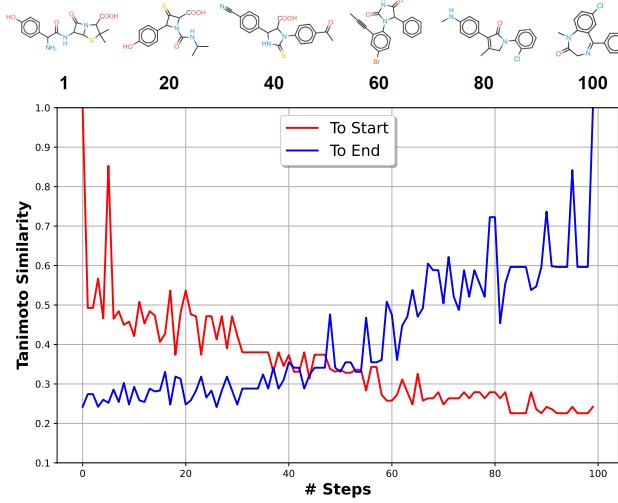

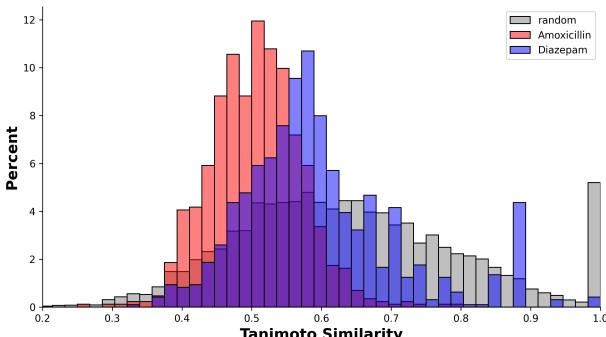

Figure 3. Density distribution of ECFP4 Tanimoto similarities of sampled molecules to their seeds. Red: similarity of 1'000 molecules to the seed Amoxicillin. Blue: similarity of 1'000 molecules to the seed Diazepam. Gray: Similarity of 10'000 sampled structures to their randomly selected seeds in the training data.

Figure 2. Linear interpolation with constant steps between two molecules in the latent space of GIRAFFE. 100 valid molecules were sampled using a temperature of 0.1. Top: Examples of sampled structures visualized with SmilesDrawer (Probst & Reymond, 2018) at given interpolation steps. Bottom: The ECFP4 Tanimoto similarity of each sampled molecule compared to the start (Amoxicillin, red) and the end (Diazepam, blue).

a linear interpolation between two example molecules in latent space using 100 equally-sized steps. Figure 2 shows the resulting similarities of sampled molecules during interpolation to both the start and end point. All 100 sampled SMILES-strings (96 unique molecules checked by InChI key) could be converted to valid molecules using RDKit (RDKit). As a comparison and as mentioned in Section 2.3.1, we performed the same experiment using the same model architecture but without variational sampling, which we call "nonVAE". Interpolating the latent space of nonVAE only decoded to 80 valid SMILES-strings (56 unique molecules).

We further evaluated the SMILES validity when randomly sampling $\sim \mathcal{N}(0, 1)$ in the latent space of GIRAFFE. Out of 10'000 randomly sampled points in latent space, 9'436 reconstructed to valid molecules (all unique, checked by InChI key) using a temperature of 0.5. This corresponds to the observed validity during training of approximately 96%. Randomly sampling the latent space of the nonVAE model the same way only decoded to 3'616 valid SMILES-strings (3'565 unique molecules). A sampling speed of around 35 SMILES per second was observed on a NVIDIA A100-SXM4-40GB GPU for random sampling with maximum 128 allowed characters.

To compare the similarity of sampled molecules to the training data, we assessed the ECFP4 Tanimoto similarity of 10'000 sampled structures compared to their "seeds", which were 10'000 random training molecules embedded using the Attentive FP encoder before sampling. The Tanimoto similarity distribution of the sampled structures to their

seeds is shown in Figure 3 with a mean of 0.64 ± 0.17 standard deviation. Depending on the seed structure, the similarity varied, which can be observed for Amoxicilin and Diazepam in Figure 3.

The distribution of physicochemical properties of randomly sampled compounds from latent space matched the one of the training data (Table A.1 and Figure A.1). The property distribution was also assessed by interpolation between two points in latent space. Figure 4 shows how the values of four properties change while linearly traversing the GIRAFFE latent space from Amoxicillin to Diazepam. The overall distribution of selected properties is visualized in Figure 5 and Figure A.6.

## 4. Discussion

With GIRAFFE we present a novel method to learn a globally applicable molecular representation. We combine the advantages of graphs as the natural molecular structure with the flexibility of SMILES-string generation and employ the VAE loss to enforce a continuous latent space. As argued by (Winter et al., 2019), a translation task is more robust than simple reconstruction, which we adopted as graph to SMILES and property translation. Our results show that the learned representation is robust for sampling novel molecules that are similar to the training data, and useful to successfully interpolate between seeds. The variability of the generated molecules can both be steered by sampling around a point of interest in latent space (*i.e.*, a molecule of interest), or by using higher temperature values for the decoder LSTM. In addition, the same representation shows successful results for QSAR tasks, enabling global applications like clustering, QSAR, virtual screening and *de novo* molecular design all in one. We argue that the relevance of the GIRAFFE latent space for

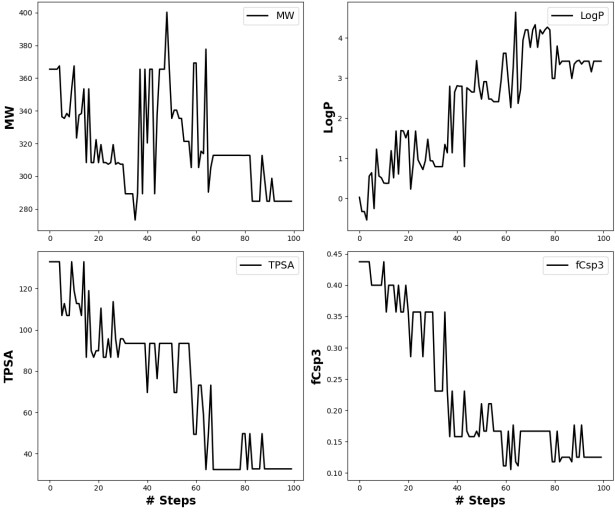

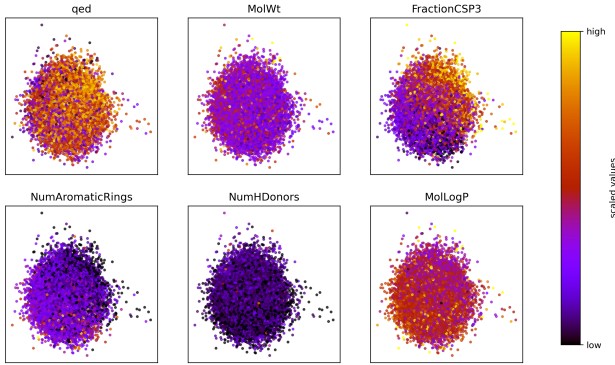

*Figure 5.* Visualization of the latent space of 25'600 random molecules from the training set embedded with GIRAFFE, using a principal component analysis for dimensionality reduction and selected scaled RDKit properties for coloring. The same visualization reduced using t-distributed Stochastic Neighbor Embedding (tSNE) is shown in Figure A.6.

*Figure 4.* Changing values of molecular properties relevant for drug discovery during a linear interpolation between two points in latent space. The start and end points are the same as in Figure 2.

QSAR applications is achieved by making use of the readily available priors in the training data with a property prediction task.

Our linear interpolation and sampling experiments have shown that $\beta$-VAE loss helps to disentangle the latent space compared to the nonVAE model. (Jin et al., 2018) argue that using SMILES prevents generative VAEs from learning smooth molecular embeddings, which we disprove in this work, as the validity of our randomly sampled molecules is the same as theirs. Also, no reinforcement learning was needed to get a high fraction of valid molecules with our approach (Blaschke et al., 2020).

To mitigate the issue of posterior collapse, where the model underutilizes the latent space, we implement a cyclical annealing schedule for the factor $\beta$ weighting $\mathcal{L}_{KLD}$ (Eq. 1). Cyclical annealing has been shown to be beneficial over monotonic annealing (Fu et al., 2019), which we could confirm in our case with a sigmoid annealing schedule.

We did not employ fine-tuning on the benchmark datasets, as we want to obtain a global representation applicable for multiple challenges and endpoints, including molecular design. Even though GIRAFFE did not outperform existing learned representations in the presented benchmarks, it outperforms ECFP4 fingerprints and scaled RDKit descriptors.

## 5. Conclusion and Outlook

With GIRAFFE, we showed that it is possible to obtain a smooth latent space representation by using a VAE with GNN encoder and LSTM decoder. The obtained latent space can be traversed or randomly sampled to recreate SMILES-strings with high validity and similarity to the training data, and is well performing for QSAR and drug design tasks at the same time. Still, more work is needed to find a molecular representation that works satisfactorily well on predictive tasks important in drug discovery (Dias et al., 2023). We will continue to train and evaluate our GIRAFFE model using actual assay readouts of compounds on biological targets, cells or from physicochemical end points to see if this further improves the performance of the learned representation, potentially also employing contrastive learning. Finally, we are looking forward to expanding this approach to property-, similarity-, docking- or scaffold-constrained generation approaches with direct impact on drug discovery projects.

## Data and Code Availability

The code used to train the here presented models together with the model weights of the GIRAFFE model is made available in the supplementary information as well as on https://github.com/alexarnimueller/giraffe. The dataset with 10M molecules from PubChem will be made available upon request.

## Acknowledgements

We would like to thank all reviewers who gave useful comments and thereby helped to improve the manuscript. We further thank Eugen Eirich and the Roche SMDA network for their ideas, feedback and critical discussions. Finally, we are indebted to the communities behind the multiple open-source software packages on which this research depends.

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

# A. Appendix

*Table A.1.* Value distribution comparing training data and 25'600 sampled molecules for selected calculated properties.

| PROPERTY | TRAINING DATA | GIRAFFE SAMPLED |
|---|---|---|
| LOGP | 3.37 ± 1.34 | 3.20 ± 1.85 |
| MOLWEIGHT | 365 ± 133 | 359 ± 113 |
| FCSP3 | 0.42 ± 0.24 | 0.42 ± 0.21 |
| NR. HBD | 1.34 ± 1.23 | 1.36 ± 1.11 |
| AROM. RINGS | 1.96 ± 1.46 | 1.87 ± 1.12 |
| QED | 0.59 ± 0.23 | 0.59 ± 0.21 |

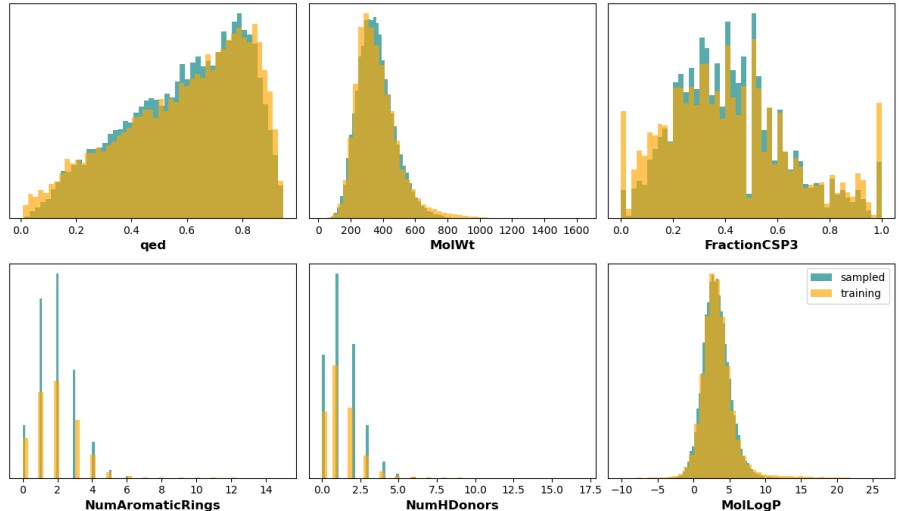

*Figure A.1.* Histograms of properties presented in Table A.1 of the training data (yellow) and randomly sampled 25'000 molecules (teal). The y-axis describes the relative frequency.

*Figure A.2.* Example molecules decoded from randomly sampled points in GIRAFFE latent space.

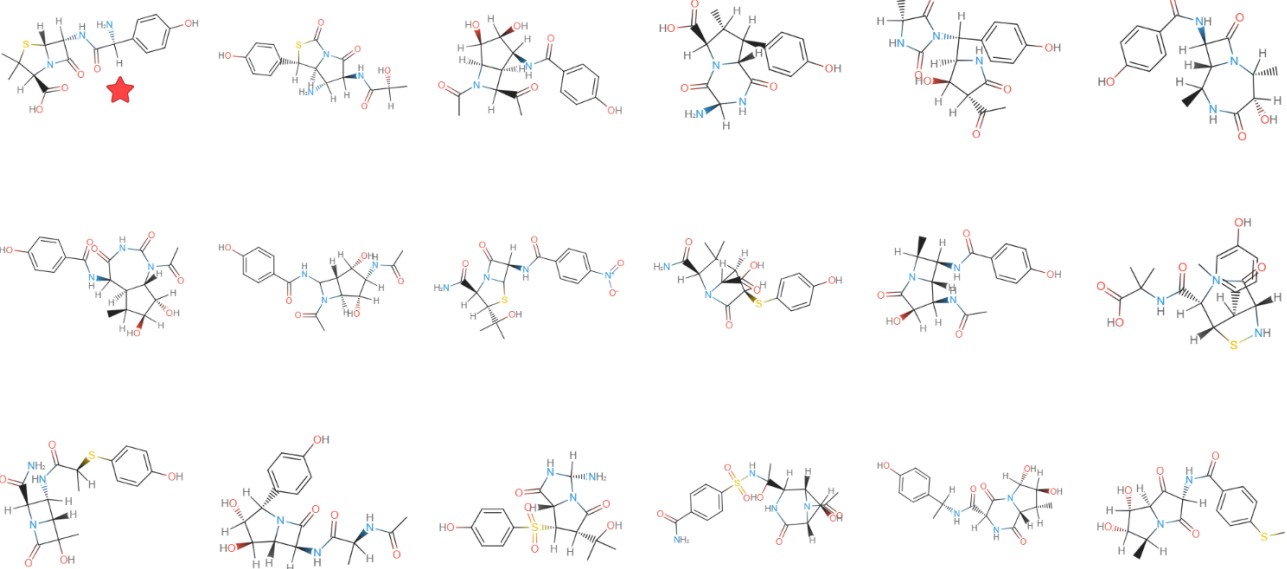

*Figure A.3.* Example molecules randomly sampled in proximity of Amoxicillin (marked by red star).

*Figure A.4.* Example molecules randomly sampled in proximity of Diazepam (marked by red star).

*Table A.2.* List of one-hot encoded atom and bond features to describe the vertices and edges of the input graph.

|  | Features | Nr. Features |
|---|---|---|
| Atom Features | atom type: C, N, O, S, P, F, Cl, Br, I, B, Si, other; degree: 0, 1, 2, 3, 4, 5, 6, other; charge, has radical electrons; hypbridization: sp, sp2, sp3, sp3d, sp3d2, other; aromatic; total Nr. of hydrogens: 0, 1, 2, 3, other; chirality: R, S, possible | 32 |
| Bond Features | bond type: single, double, triple, aromatic, conjugated, ring; stereo: none, any, Z, E | 10 |

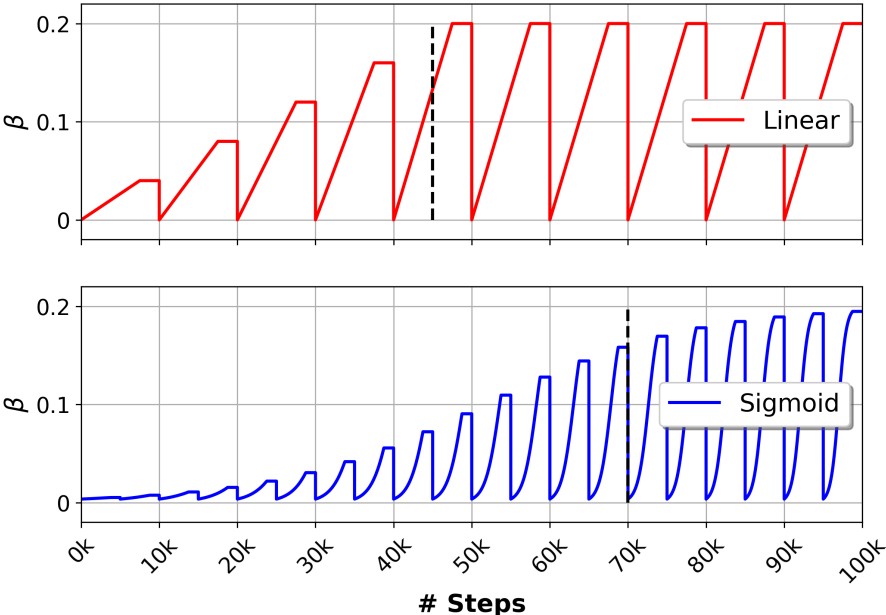

*Figure A.5.* Two best performing cyclical annealing strategies for $\beta$ values during training. Top (red): Linear increase over 4 cycles with cycle sizes of 10'000 steps with 7'500 increasing and 2'500 constant steps. Bottom (blue): Sigmoidal increase over 20 cycles with cycle sizes of 5'000 steps with 3'750 increasing and 1'250 constant steps. Both strategies were allowed to reach a maximum $\beta$ value of 0.2, and performed best in the tested benchmarks at the step indicated by a dashed line.

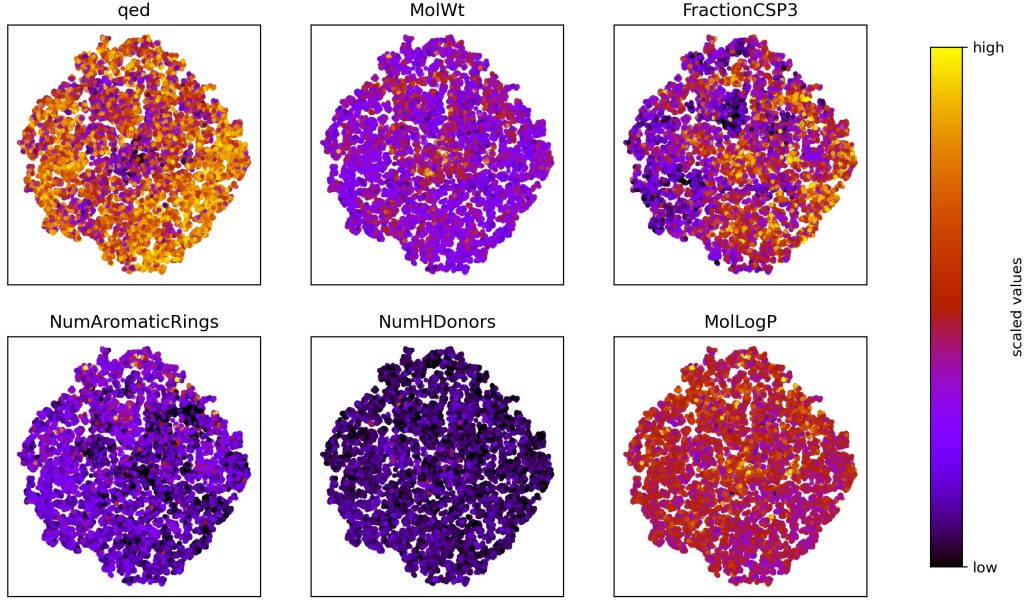

*Figure A.6.* Visualization of the latent space of 25'600 random molecules from the training set embedded with GIRAFFE, using tSNE (Van der Maaten & Hinton, 2008) for dimensionality reduction and selected scaled RDKit properties for coloring. A PCA of the same data is shown in Figure 5.