# OpenReview forum: "Combining Graph Attention and Recurrent Neural Networks in a Variational Autoencoder for Molecular Representation Learning and Drug Design"
_ICML.cc/2024/Workshop/ML4LMS — ML4LMS Poster_

### Official Review · Reviewer_jGA6 · 2024-05-30
**Maybe present some outcomes of different annealing approaches !**

**Rating:** 6
**Confidence:** 3

**Review:**

The manuscript introduces an approach by combining graph attention neural networks with RNN, specifically LSTM, within a VAE. This fusion leverages the strengths of both architectures for molecular representation learning, which is a relatively new and promising. The proposed method aims to capture the properties of molecular structures effectively, enabling more efficient encoding and flexible decoding processes. Authors demonstrated competitive performance in quantitative structure-activity relationship (QSAR) benchmarks, as well as high validity and drug-likeness of randomly sampled molecules. Moreover, the approach shows robustness for linear latent space interpolation between two molecules.

Further experimentation is required as authors discussed in Conclusion and Outlook section which will make the work more credible to other researchers.

Overall, promising results in benchmarks and downstream tasks, and a roadmap for future research suggests that the work described is indeed unique in the field of molecular representation learning and has the potential to contribute significantly to chemical sciences and drug discovery.

Some suggestions that authors might experiment in the future:
1. Dynamic annealing schedules, e.g. cosine annealing or exponential annealing.
2. Multi stage annealing.
3. Optimize the annealing schedule along with other hyperparameters of the model. Involve techniques like RL based approaches or Bayesian optimization to automatically search for the optimal annealing schedule during training.
4.  Train multiple VAE with different annealing schedules or hyperparameters and combining their predictions through ensemble methods.

---

### Official Review · Reviewer_7LXZ · 2024-06-05
**Molecular VAE with some comparison to prior art and good description of AI4Chemistry literature**

**Rating:** 7
**Confidence:** 4

**Review:**

Originality: This work incorporates a combination of well-known techniques, and it is very clear how this work is different from prior research. Most of the prior work has received a good citation.

Quality: The submission is technically not very sound, and claims are not well supported, but a lot of emphasis is put on showcasing model architectures and training. This is a work in progress because an extension to evaluation metrics will be nice to see, and other data sets are required.

Clarity: The paper has a good flow. The exact results are reproducible.

Significance: I truly believe that the results here can be important. The ideas in this paper will likely be used and expanded upon by others.

Strengths: Some comparison to prior art is mentioned, and a good literature overview of AI4Chemistry is showcased.

Weaknesses: More comparison to prior art will be nice to have, especially in the random sampling/generation study. There should be so many generative models based on the VAE architecture for molecules (i.e., G'omez-Bombarelli et al., 2018), and close to no comparison has been shown to those (both metric and architecture-wise).

---

### Official Review · Reviewer_TGDR · 2024-06-12
**Good paper on molecular representation learning**

**Rating:** 7
**Confidence:** 3

**Review:**

The authors propose a VAE with a GNN encoder and a LSTM decoder that creates a continuous latent space representation of molecules. This allows for interpolation between molecules as well as generation of new candidates via SMILES strings. The model also includes an element for the evaluation of molecular properties.
The paper is well-written and easy to follow. The approach overcomes earlier issues in the creation of a continuous latent space. Results indicate that sampled molecules are mostly valid.

Questions and comments:
How do you evaluate the MLP for property prediction? Are the predicted properties for the newly generated molecules plausible?